# Upconversion particle-assisted NIR polymerization enables microdomain gradient photopolymerization at inter-particulate length scale

Peng Hu[1], Hang Xu[1], Yue Pan[1], Xinxin Sang[1,2] & Ren Liu [1,2] ✉

High crosslinking and low shrinkage stress are difficult to reconcile in the preparation of performance-enhancing photopolymer materials. Here we report the unique mechanism of upconversion particles-assisted NIR polymerization (UCAP) in reducing shrinkage stress and enhancing mechanical properties of cured materials. The excited upconversion particle emit UV-vis light with gradient intensity to the surroundings, forming a domain-limited gradient photopolymerization centered on the particle, and the photopolymer grows within this domain. The curing system remains fluid until the percolated photopolymer network is formed and starts gelation at high functional group conversion, with most of the shrinkage stresses generated by the crosslinking reaction having been released prior to gelation. Longer exposures after gelation contribute to a homogeneous solidification of cured material, and polymer materials cured by UCAP exhibit high gel point conversion, low shrinkage stress and strong mechanical properties than those cured by conventional UV polymerization techniques.

Photopolymerization technology has received remarkable interest due to its full temporal and spatial resolution advantages allowing flexible and efficient fabrication of functional polymers in mild conditions[1-4], and the variety of monomers and oligomers offers a multitude of possibilities for modulating polymer properties[5-7]. Despite these advantages, high shrinkage stress in curing process and poor penetration of UV light limit the further improvements in ultimate mechanical properties. Firstly, the curing system undergoes an auto-acceleration after initiation, accompanied by rapid crosslinking of the monomers, and then fast turns into a glassy where the frozen molecules are unable to move, resulting in the accumulation of shrinkage stress and defects in material properties[8]. Optimizations of curing conditions and parameters, or adding fillers to the curing system can reduce shrinkage stresses to some extent[9-11]. In addition, the light source irradiates on the sample surface in a two-dimensional plane during the curing process and the light intensity is distributed in a gradient decrease in the thickness direction, resulting in heterogeneity of conversion in the cured material[10,12]. A feasible solution is to introduce photoinitiation system with long-wavelength absorption due to the greater penetration nature of long wavelength light sources[13]. NIR induced photopolymerization represents the latest development in this field[14-18], and in particular UCAP is compatible with the vast majority of conventional UV-curable formulations[19-21], shows attractive application potential in the field of deep light curing, 3D printing and optical anti-counterfeiting, etc.[22-25].

Lanthanide-doped upconversion particles (UCPs) as internal lamps disperse inside the formulation and convert incident NIR to UV-vis light through anti-Stokes process[26,27], and then NIR photopolymerization can be achieved by matching the UCPs fluorescence to the absorption band of conventional UV initiators[28,29]. As an emerging photopolymerization technology, quality optimization of materials prepared by UCAP is often based on trial, this can be improved by

[1]International Research Center for Photoresponsive Molecules and Materials, Jiangnan University, Wuxi 214122 Jiangsu, PR China. [2]Key Laboratory of Synthetic and Biological Colloids, Ministry of Education, Jiangnan University, Wuxi 214122 Jiangsu, PR China. ✉e-mail: liuren@jiangnan.edu.cn

developing better understanding of the polymerization process. In our previous work, we systematically investigated several factors affecting the initiation process, including the efficiency of various commercial UV photoinitiators in utilizing the fluorescence from UCPs, and an evaluation method has been developed to help select suitable photoinitiators[21]. To further improve the initiation efficiency, coumarin-based photoinitiators with high spectral match were grafted onto the UCPs surface and activated by a Förster resonance energy transfer mechanism[30]. In addition, the thermal effect of the laser and the extinction effect of the filler was also considered in relation to the initiation efficiency[28,31].

In UCAP, despite the addition of the UCPs as up-conversion emitters in the initiation process, following chain growth and polymer formation appear to be identical to conventional UV photo-polymerization. However, it is neglected that the light sources used in UCAP are micro-lamps dispersed in the resin matrix rather than conventional planar UV sources. The effect of irradiation condition changes on the polymerization behavior and the ultimate properties of cured material are still mysterious and unknown. The current work was inspired by an unexpected experimental finding that the final performance of UCAP-cured materials was higher than that of conventional UV-cured materials[31], but the existing studies of UCAP are mostly based on macro-scale characterization methods, which cannot reveal the reasons for the enhanced performance. Although Khaydukov et al. observed the curing process around UCPs in micro-scale while developing the NIR laser scanning photolithography based on UCAP[32], to date, detailed studies have not been extended to consider the curing behavior at inter-particulate length scale, which stirred our desire to conduct further explorations.

In this article, we investigated the curing mechanism of UCAP at the micron scale and its effect on the material properties (Fig. 1). Firstly, we visualize the growth of polymer microspheres around UCPs over time by performing polymerization reactions in resin matrix with extremely dilute UCPs concentrations, and the distribution of the double bond conversion around single UCPs and between two particles were in situ detected by confocal Raman microscopy. Furthermore, NIR photorheology experiments were performed to investigate the effects of light intensity and UCPs concentration on time to gel point and conversion at gel point. Finally, shrinkage stress and ultimate properties of cured material under UCAP and UV curing conditions were measured to clarify the effect of curing mechanisms.

## Results and discussion
### Polymerization reaction around UCPs
In order to obtain a high initiation efficiency during the UCAP process, the fluorescent bands of the UCPs should overlap as much as possible with the photoinitiator absorption bands. Thulium-doped UCPs are ideal candidates for UCAP due to their UV−visible emission band which can overlap with some of the absorption bands of conventional photoinitiators. In addition, micron-scale UCPs exhibit higher upconversion efficiency and emission intensity due to the fewer defects on their surface compared to the nanoscale UCPs[33]. In this experiment, rod-like UCPs with a diameter of ~5 μm hexagonal bottom and a length of 14 μm (inset in Fig. 2a or Supplementary Fig. 1) were synthesized through hydrothermal method, corresponding upconversion fluorescence emission spectrum under the excitation of 980 nm laser was shown in Fig. 2a, and the integral of UCPs emission intensity between 330−375 nm vs. the excitation intensity was exhibited in Fig. 2b. The emission band at 350 nm can partially overlap with the absorption band of photoinitiator BAPO, which is suitable for photopolymerization experiments.

The UCAP process is realized by the simultaneous up-conversion emission of numerous UCPs dispersed in the resin matrix under NIR excitation, it is essential to observe the polymerization process around a single UCPs to reveal its curing mechanism. To achieve this, polymerization experiments were performed in a photocurable resin (BPA10EODMA) containing a very low UCPs concentration of 0.1 wt%, the NIR laser intensity was controlled at 25 W/cm², then the polymer shell growth around the surface of single particle can be observed through optical microscopy or SEM, see Fig. 2c, the bright-field images were presented where individual UCPs and polymer shells can be clearly distinguished, and corresponding SEM images were also provided to observe the three-dimensional morphology. It is clear to see that as the irradiation time increases, the polymer shell around single UCPs grows thicker and finally forms a perfect microsphere. When slightly increase the UCPs concentration to 0.5 wt% and simultaneously increase the irradiation time, a percolated microsphere network can be observed through interpenetration between two or more polymer microspheres (Fig. 2d), which indicates that the overall curing of the material is obtained through the connection of numerous polymer microspheres.

### Conversion distribution at inter-particulate length scale
Although the formation of polymer microspheres around UCPs has been clearly observed in dilute solutions, the distribution of conversion in this area and at interparticle distances is still unknown. Confocal Raman Microscopy (CRM) as a non-invasive in situ monitoring technique, was used here to characterize double bond conversion surrounding UCPs. Figure 3a represents a 50 × 50 μm bright-field image of an individual UCPs dispersed in resin matrix and the distribution of double bond conversion over the same 2D area, color scale is used to highlight relatively high (blue) and low (red) value areas. It

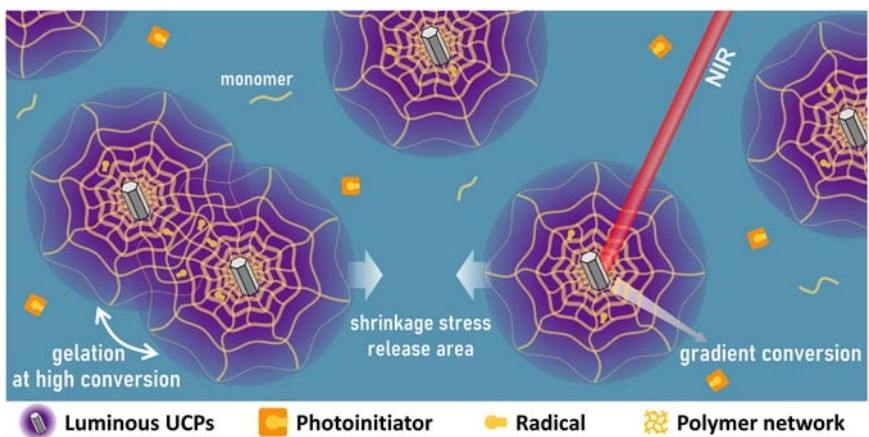

**Fig. 1 | Upconversion particles-assisted NIR polymerization process.** Schematic illustration of microdomain gradient photopolymerization and stress release mechanism.

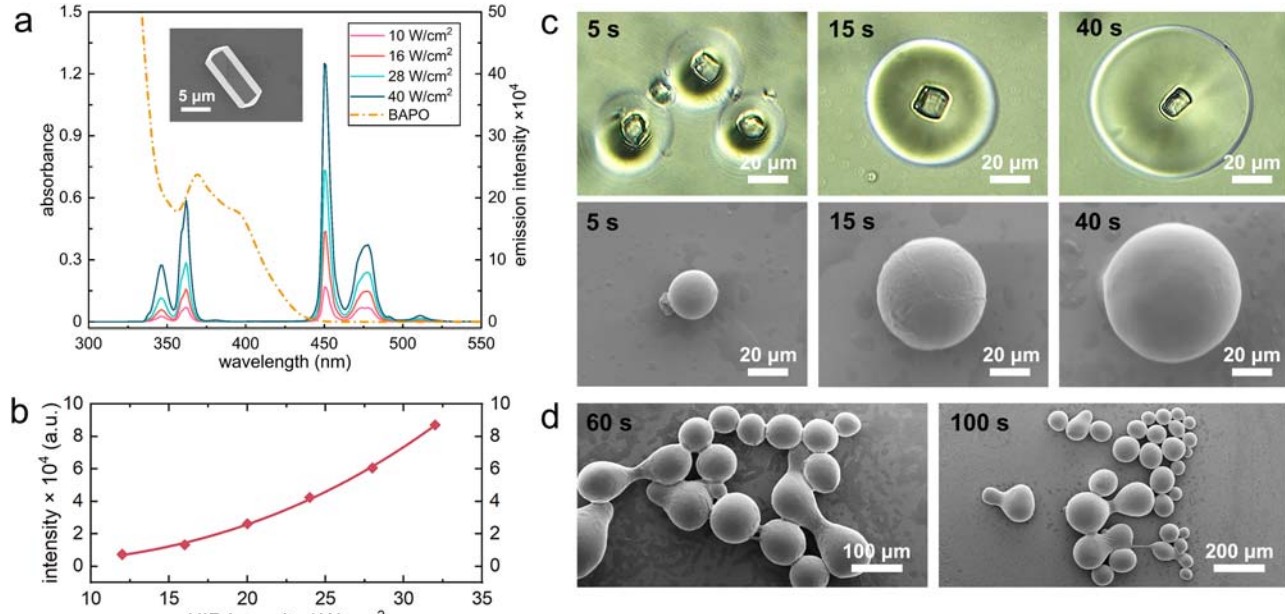

**Fig. 2 | Fluorescence properties of UCPs and polymerization reaction around UCPs. a** Absorption spectrum of the photoinitiator BAPO in acetonitrile and the emission spectrum of the UCPs under gradient laser intensity excitation, the inset is an SEM image of the UCPs used in this experiment. **b** Integral of UCPs emission bands between 330–375 nm vs. the excitation intensity at 980 nm. **c** Morphology of polymer microspheres around individual UCPs cured at different irradiation times (0.1 wt% UCPs, 25 W/cm² NIR intensity). **d** Morphology of linked polymer microspheres cured at higher UCPs concentrations (0.5 wt% UCPs, 25 W/cm² laser intensity).

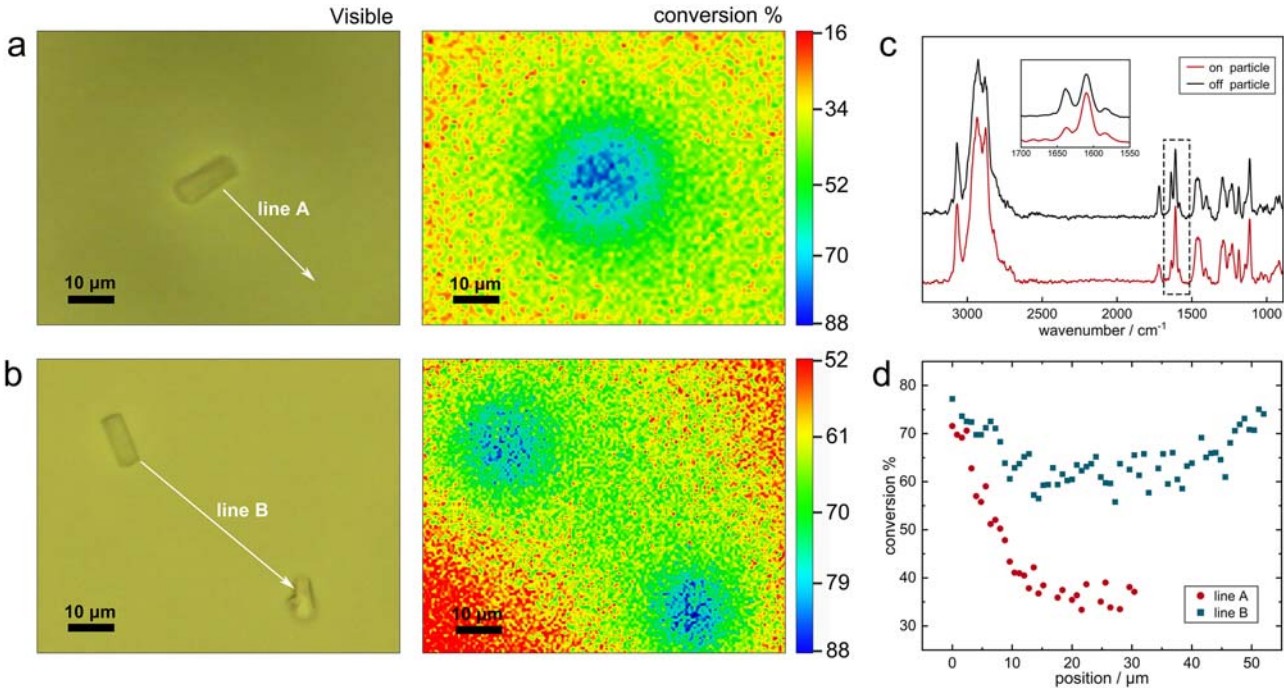

**Fig. 3 | Distribution of double-bond conversion around single and two UCPs. a** Bright-field image of an individual UCPs in curing system and the distribution of double bond conversion. **b** Bright-field image of two separated UCPs dispersed in curing system and the distribution of double bond conversion. **c** Corresponding Raman spectra at the particle center (on particle) and away from the particle (off particle). **d** Double bond conversion distribution of line A and line B located in (**a**) and (**b**).

can be seen that the process of UCAP is a microdomain-gradient photopolymerization, the degree of conversion decreases in a gradient from the center of the UCPs to the resin matrix, the spectral evolution of the double bond band at the particle center and away from the particle is given in Fig. 3c, weaker alkene peak (1638 cm⁻¹) is observed

on the particle and increase at off particle area. A more detailed distance-conversion profile is plotted in Fig. 3d (line A). This heterogeneity of conversion derives from the light intensity gradient of UCPs fluorescence in resin matrix, the UCPs as internal UV lamps emit photons to the surrounding, resulting in high light intensity in the near

and low light intensity in the far area, and finally form conversion gradient consistent with the light intensity gradient. The conversion distribution map between two UCPs is also characterized and shown in Fig. 3b, the conversion between two particles is significantly higher than the area where no overlaps, detailed conversion trend between two particles is shown in Fig. 3d (line B), where the distance between the two particles is ~50 μm and the conversion is lowest at the midpoint of the line B (distance at 25 μm) with a value of 60%, significantly higher than the 35% conversion at 25 μm distance from isolated particle (line A), indicates that the overlap of the irradiated area from two particles increases the final conversion. As the exposure time increases and the irradiation dose reaches saturation, the monomers in the low conversion area will further react completely and finally the system shows a homogeneous high conversion distribution inside (see Supplementary Fig. 2).

## Conversion delaying at gel point

The overall solidification of liquid resin origin from the connection of numerous polymer microspheres to form a continuous polymer structure. Prior to this, the curing system maintains fluidity until the connection formed and start to gelation, as shown in Fig. 4a. The main influencing factors of this process include the growth rate of polymer

shell around UCPs, which is determined by light intensity and dose, and the distance between adjacent particles, which is determined by UCPs concentration. Photorheology has been reported for real-time monitoring of the gelation process of photopolymerization reaction[34,35]. Here, a modified ATR-IR-photorheology device was used to monitor the gelation process of curing system under different NIR light intensity and UCPs concentrations, see Fig. 4b, the irradiation source (980 nm laser) was turned on at 30 s, the polymerization reaction occurred after a short induction period, the storage modulus G' and loss modulus G" began to rise. The intersection of G' and G" is defined as the gel point which indicates the transition from liquid resin to solid polymer, the conversion at gel point was provided via monitoring the signal decreasing of acrylate double bond. The distance between adjacent particles will shorten with the increase of UCPs concentration, and the time required for the interpenetration between polymer microspheres under constant light irradiation will also be shorter, resulting in the shortening of the gelation time, and short gelation time lead to lower irradiation doses, which in turn cause the gel point conversion to decrease as the concentration of UCPs increases, exhibit in Fig. 4c. Typically, the luminescence intensity and radiation range of the UCPs increases with laser intensity (Fig. 2b), which will accelerates the growth of the polymer microspheres around

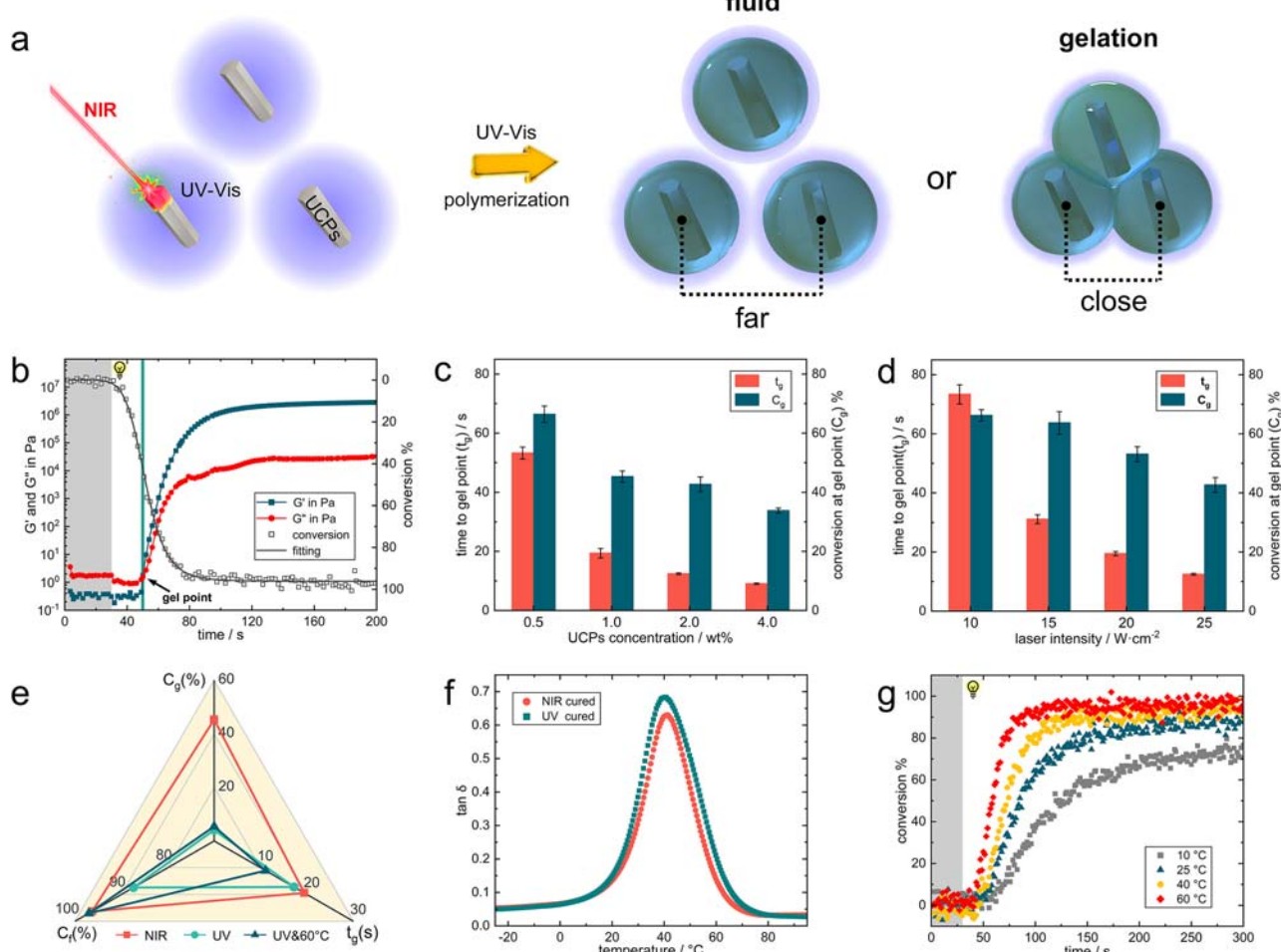

**Fig. 4 | Gel behavior of upconversion particles-assisted NIR polymerization.**
**a** Schematic presentation of the gelation process. **b** Example of photorheology test for gel point detection using 1 wt% UCPs formulation (NIR intensity 25 W/cm²). **c** Time to gel point ($t_g$) and gel point conversion ($C_g$) at different UCPs concentrations (25 W/cm²) and at (**d**) different NIR intensities (2 wt% UCPs). The error bars represent the standard deviations.
**e** Comparison of gel point conversion ($C_g$), time to gel point ($t_g$) and final

conversion ($C_f$) under three different curing conditions: NIR, UV and UV coupled with heating (NIR 25 W/cm², 1 wt% UCPs, UV 8 mW/cm², heating to 60 °C). **f** Tan($\delta$) as a function of temperature for a resin (1 wt% UCPs) cured under NIR and UV respectively (NIR 25 W/cm², UV 8 mW/cm² at 25 °C).
**g** Conversion-time profiles at different ambient temperatures under UV irradiation (8 mW/cm²).

**Table 1 | Data summary of time to gel point, gel point conversion under different curing conditions**

| UCPs(wt%)[a] | $t_g$(s) | $C_g$ (%) | $C_f$ (%) | $P_{NIR}$(W/cm²)[b] | $t_g$ (s) | $C_g$ (%) | $C_f$ (%) |
|---|---|---|---|---|---|---|---|
| 0.5 | 53 ± 2 | 66 ± 3 | 96 ± 3 | 10 | 73 ± 3 | 66 ± 2 | 91 ± 1 |
| 1.0 | 19 ± 2 | 45 ± 2 | 97 ± 1 | 15 | 31 ± 2 | 63 ± 4 | 94 ± 2 |
| 2.0 | 12 ± 1 | 43 ± 2 | 98 ± 1 | 20 | 20 ± 1 | 53 ± 2 | 95 ± 1 |
| 4.0 | 9 ± 2 | 34 ± 1 | 98 ± 1 | 25 | 12 ± 1 | 43 ± 2 | 98 ± 1 |
| UV[c] | 18 ± 1 | 4 ± 1 | 86 ± 2 | UV&60 °C[d] | 12 ± 1 | 5 ± 1 | 97 ± 1 |

$t_g$ time to gel point, $C_g$ conversion at gel point, $C_f$ final conversion, $P_{NIR}$ laser intensity of NIR.
[a] NIR intensity is 25 W/cm².
[b] UCPs concentration is 1 wt%.
[c] UV irradiation at 8 mW/cm².
[d] UV irradiation (8 mW/cm²) coupled with 60 °C heating.

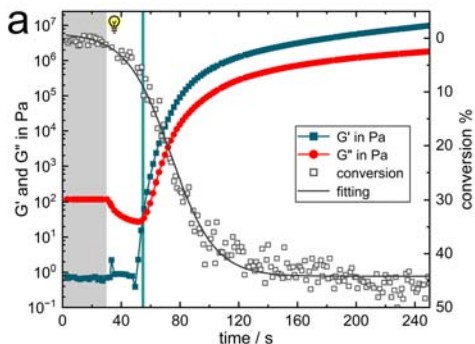
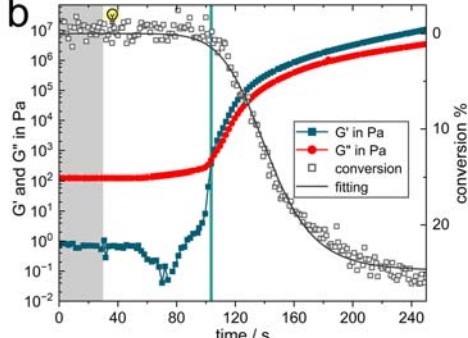

**Fig. 5 | Gel behavior of high viscosity resin systems. a** Photorheology test for gel point detection in difunctional bisphenol A epoxy acrylate formulation (RY1101/ TMPTA = 7/3, 1 wt% UCPs, 20 W/cm² NIR intensity) under NIR irradiation and under (**b**) UV irradiation (2 mW/cm² UV intensity).

UCPs and reduces the interpenetration time between adjacent UCPs, exhibiting a decrease in gel time with increasing laser power (Fig. 4d), and the decrease of gel point conversion may be caused by the increase of light intensity in the overlapping part of adjacent particle luminescence domain, the light intensity reaches the polymerization threshold after superposition, and the system gels in a very short time, showing the decrease of conversion, more detailed data can be found in Table 1.

Tests were also performed in UV photopolymerization to clarify the mechanism of gel point conversion delay in UCAP. Low intensity UV light (8 mW/cm²) was used as the radiation source to simulate the weak fluorescence emission of UCPs, and the curing system was physically heated to 60 °C during the measurement process to simulate the optical heating of the NIR laser, the heating temperature for UV curing is determined by monitoring the temperature evolution of monomer under NIR irradiation (see Supplementary Fig. 3), the equilibrium temperature of the monomer (without initiator) containing 1 wt% UCPs after 25 W/cm² NIR irradiation will reach to 60 °C. Figure 4e shows the data of time to gel point ($t_g$), conversion at gel point ($C_g$) and final conversion ($C_f$) under three different curing conditions, i.e., NIR, UV, and UV coupling heating. Increasing the temperature of the UV curing system to 60 °C resulted in a slight reduction in $t_g$ from 17.2 to 11 s, Supplementary Fig. 4 shows the $t_g$ and $C_g$ data for UV curing systems at different temperatures. However, there was no significant change in $C_g$, which was less than 10% for both the ambient and heated UV curing systems (4% for UV and 5% for UV&80 °C), while the NIR system showed a high $C_g$ of 66%, indicating that the thermal effect of NIR was not the main factor causing the delaying in $C_g$ but rather the percolating of the polymer microspheres centered on the UCPs was. Increasing the temperature of the UV curing system showed a positive effect on increasing the final conversion from 87% in the ambient system (25 °C) to 97% in the 60 °C heated system. The low glass-transition temperature ($T_g$) can be used to explain the increase of final conversion. The $T_g$ of the UV-cured and NIR-cured samples were tested by DMTA and were close to each other at ~40 °C (final conversion of UV and NIR-cured samples were 86% and 97% respectively), shown in Fig. 4f. In the UCAP process the system is heated by the NIR laser and the temperature is maintained at ~60 °C above the $T_g$[36], which helps the trapped chains react further since the network mobility is increased during the optical heating, thus the final conversion of UV-cured material also increases with the ambient temperature, see Fig. 4g.

An increase in system viscosity tends to accelerate the gelation process, and the conversion of a small number of functional groups allows for a rapid increase in viscosity and hence gelation. Therefore, the photorheology experiments were carried out with a commonly used epoxy acrylic resin system and the results are shown in Fig. 5a, b. The gel point conversion of the system under NIR curing was around 8%. In contrast, in UV curing system the gel point conversion is less than 2%. All evidence proves that the specific curing mechanism of UCAP is effective in increasing the gel point conversion.

## Reduction of shrinkage stress in polymerization

The delay in gel point conversion will contribute to the reduction of shrinkage stress in conventional photopolymerization systems[8]. Photorheology analysis has been proven to be a viable method for evaluating the shrinkage stress by monitoring the evolution of normal stress during curing tests[35], but in NIR curing experiments, the sample and test rotor exhibit thermal expansion under laser optical heating, resulting in an increase in normal stress in the positive direction at the end of curing (Supplementary Fig. 5). Therefore, a new evaluation strategy was devised as shown in Fig. 6a, where the shrinkage stress of the sample can be assessed by comparing the degree of deformation of the substrate, and the resin used here has been replaced with a combination of greater viscosity formulation (RY1101/TMPTA = 7/3),

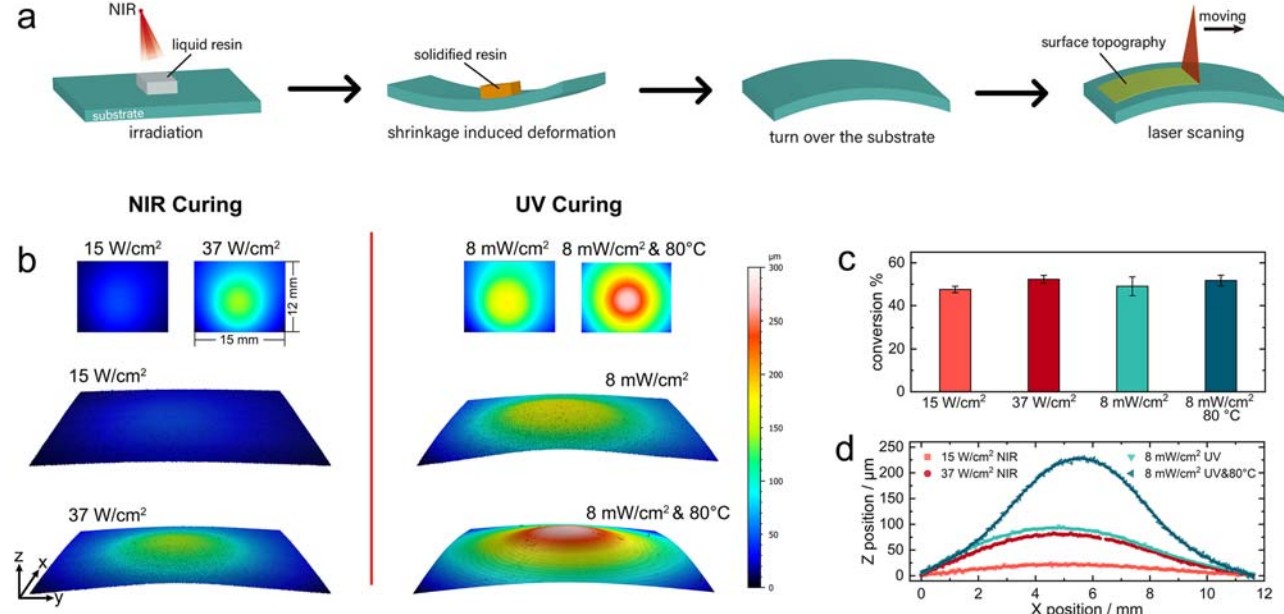

**Fig. 6 | Comparison of shrinkage stress between NIR and UV-cured materials. a** Schematic presentation of detecting shrinkage stress. **b** Comparison of stress induced substrate deformation under NIR and UV curing conditions respectively. **c** Final conversion of cured materials under different curing conditions, the error bars represent the standard deviations. **d** Y-axis contour lines through the highest point of deformed substrates in (**b**) (1 wt% UCPs, 1 wt% BAPO, RY1101/TMPTA = 7/3).

allowing for more significant deformation. The topography of the substrate at the bottom was tested at different NIR laser intensities and, for comparison, samples prepared by UV and UV combined with NIR heating were also tested (Fig. 6b), by adjusting the NIR and UV light intensities separately, the conversion of sample under 15 W/cm² NIR irradiation was close to that at 8 mW/cm² UV irradiation, and the conversion under 37 W/cm² NIR irradiation was close to that at 8 mW/cm² UV irradiation combined with 80 °C laser heating, see Fig. 6c. The deformation of the substrate increases slightly with the increase of the NIR light intensity. At a laser power of 15 W/cm² there is almost no deformation on the substrate surface, and when the light intensity is increased to 37 W/cm², the difference between the highest and lowest point of the deformed substrate is only 140 μm. The deformation of the UV-cured samples was significantly greater than that of the NIR samples, the UV-cured samples combined with heating showed the greatest deformation with a height difference of up to 300 μm and exhibited a larger area of deformation than the NIR cured samples, the corresponding Y-axis contour line through the highest point of deformation is given in Fig. 6d. The greater deformation in the UV combined heating system may be attributed to the faster polymerization rate, see Supplementary Fig. 6, where the polymerization reaction is completed more rapidly in the heating system. The system gels at a low conversion and the stresses generated by the subsequent rapid crosslinking cannot be released, leading to the accumulation of stresses in the material, and the thermal stresses in the material may also be responsible for the increased deformation[37,38]. For the NIR curing system, the polymerization rate is close to those of UV curing, see Supplementary Fig. 6, but the degree of substrate deformation is much lower than that of the UV system, indicating that the delayed gel point conversion is the key factor in reducing shrinkage stress in the system.

## Mechanical properties of photocured materials

Formulation parameters and NIR irradiation conditions affect the final conversion and mechanical properties of UCAP cured materials to a certain extent[31,39]. The shrinkage stress also affects the mechanical properties especially in the conventional UV curing process, harmful shrinkage stresses usually weaken the performance and lifetime of the material. Considering that UCAP shows great potential for reducing shrinkage stresses in materials, the mechanical properties of NIR-cured materials were therefore further tested and compared with those of UV-cured materials. Figure 7a shows the tensile strength data and final conversion data of samples obtained at different NIR curing times and, as a reference, the data for UV-cured samples are also provided. As shown in Fig. 7a, as the irradiation time increased, the conversion of the material gradually increased and eventually became greater than that of the UV-cured material, while the tensile strength of the material showed a trend of increasing and then slightly decreasing with increasing irradiation time, Fig. 7b shows a more detailed stress-strain profile. The tensile strength reaches a maximum of 70 MPa after 50 s irradiation with a final conversion of 63%, which is close to the 65% conversion of UV-cured material, but the tensile strength of the UV-cured material is only ~50 MPa. In addition, UCAP cured materials showed higher tensile modulus and tensile toughness than UV cured materials (see Supplementary Table 1). And materials cured by NIR show better impact resistance compared to UV-cured materials (see Supplementary Movie 1). Combined with the results of the shrinkage stress in the previous section, the reduction of shrinkage stress is likely to be the main factor contributing to the performance enhancement, suggesting that UCAP curing technology shows great promise for the preparation of performance-enhancing polymers.

## Methods

### Materials

Difunctional bisphenol A epoxy acrylate oligomer (RY1101), Trimethylolpropane triacrylate (TMPTA) were purchased from Jiangsu Kailin Ruiyang Chemical Co., Ltd. (China); Ethoxylated bisphenol-A dimethacrylate (BPA10EODMA) were purchased from Eternal Materials Co., Ltd (China). Silica was purchased from Evonik (AEROSIL 200); Phenyl bis(2,4,6-trimethylbenzoyl)-phosphine oxide (BAPO) was purchased from Gurun technology Co., Ltd. (China). YbCl₃·6H₂O, TmCl₃·6H₂O, NaF, and EDTA-2Na were purchased from Sigma-Aldrich. All reagents were used without further purification unless otherwise stated.

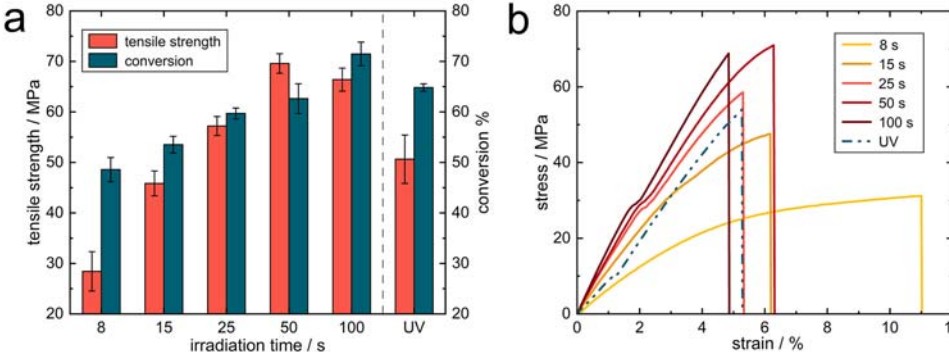

**Fig. 7 | Comparison of mechanical properties between NIR and UV-cured materials. a** Tensile strength and conversion data of NIR cured materials (UV as reference) under different irradiation time. The error bars represent the standard deviations. **b** Stress-strain profiles of cured materials. (40 W/cm² NIR intensity, 200 mW/cm² UV intensity, 2 wt% UCPs, 1 wt% BAPO, RY1101/TMPTA = 7/3).

## UCPs synthesis

UCPs were synthesized through hydrothermal method, In a typical synthesis of NaYbF₄:Tm (99.5 mol% Yb, 0.5 mol% Tm), YbCl₃·6H₂O (0.995 mmol, 385.4 mg) and TmCl₃·6H₂O (0.005 mmol, 1.9 mg) was dissolved in 20 ml deionized water. Then 10 ml EDTA-2Na aqueous solution (0.1 M) was added to form a mixture and stirred for 30 min, next 16 ml NaF aqueous solution (1 M) was added dropwise to form a colloidal mixture and the resultant solution was stirred for 1 h at room temperature. The mixture was transferred to the 50 ml teflon autoclave and heated to 180 °C. After 24 h of reaction, the mixture was naturally cooled to room temperature, the obtained precipitate was washed with alcohol (100 ml) and deionized water (100 ml), and finally dried in a drying oven to yield product 200 mg (73%).

## Polymerization in dilute UCPs formulations

Small amounts of UCPs (5 mg, 0.1 wt%) were added to the BPA10EODMA (5 g) monomer (dissolved 50 mg photoinitiator), then using a high-speed mixing machine (DAC 330-100 SE, FlackTek SpeedMixer, USA) for 5 min at 1500 rpm to give a photocurable sample. For the polymerization experiments, the sample was added to a tubular silicone mold (8 mm diameter and 4 mm deep) and then irradiated under 25 W/cm² NIR laser (FC-W-980H-50W, 980 nm, Changchun New Industries Optoelectronics Technology Co., Ltd, 1 × 1 cm square light spot). The irradiated sample was poured into a large amount of dichloromethane (5 ml) and washed three times, the polymer spheres will float on the solution surface and can be collected and placed on silicon wafer for SEM observation and in pure monomer for optical microscopy observation.

## Emission spectra

Photoluminescence spectra of UCPs were recorded by a modified fluorescence spectrophotometer (FS5, Edinburgh instrument company, UK) with a 980 nm CW laser excitation (FC-W-980H-50W, Changchun New Industries Optoelectronics Technology Co., Ltd, China). To avoid sedimentation, UCPs with a concentration of 1 wt% were dispersed in BPA10EODMA monomer and then placed in a 1 cm square cuvette for testing.

## Confocal Raman microscopy

The samples for CRM were prepared by mixing 1 wt% photoinitiator and 0.5 wt% UCPs into BPA10EODMA matrix, then using a high-speed mixing machine (DAC 330-100 SE, FlackTek SpeedMixer, USA) for 5 min at 1500 rpm. A 300 μm thick silicone mold with a hole (diameter 8 mm) in the middle was placed on the coverslip, the hole was filled with sample and covered with another coverslip to avoid oxygen inhibition, then exposed to 980 nm laser (25 W/cm²) to obtain Raman testing samples. Raman spectra and mapping images were acquired on the DXR3xi Confocal Raman microscope system (Thermo Fisher, USA) with a 532 nm laser and a ×50 objective. Each area was scanned in 0.8 μm step and corresponding conversion profiles were plotted by calculating the peak area ratio of 1638 cm⁻¹/1600 cm⁻¹ compared to uncured resin.

## Measurement setup for RT-IR-photorheology

Real-time FTIR rheological analysis was performed with NIR light irradiation employing a setup coupled with ATR-FTIR (Nicolet iS10 series, Thermo Fisher) and rheometer (HAAKE MARS60 equipped with Rheonaut annex, Thermo Fisher). A specific sample volume (150 μl) was placed at the center of the ATR plate, and the measurements were conducted at a defined temperature (25 °C) and sample thickness (300 μm). The formulations (BPA10EODMA, 1 wt% BAPO) were sheared by using oscillation time sweep mode with 1 Hz oscillation frequency and 5% strain. During the measurements, the storage modulus and loss modulus were recorded. The intensity of the laser was measured using a calibrated fiber optic spectrometer (LP100, Changchun New Industries Optoelectronics Technology Co., Ltd). The evolution of double-bond content was continuously monitored by the decrease of the area of 1638 cm⁻¹ band, and the benzene peak at 1600 cm⁻¹ was used as the internal reference. The double-bond conversions (DC%) were calculated from Eq. (1) below:

$$DC(\%) = \left(1 - \frac{A_{X,t}}{A_{X,0}} \cdot \frac{A_{ST,0}}{A_{ST,t}}\right) \times 100 \tag{1}$$

where $A_0$ and $A_t$ represent the area of the IR absorption peak of functional group before irradiation and after $t$ time. The subscripts $ST$ and $X$ represent the reference peak and the double bond peak.

## Measurement of substrate deformation caused by shrinkage stress

A silicone pad (12 mm diameter, 1 mm thick) with a circular hole (8 mm diameter) in the center was attached to a square glass sheet (24 × 24 mm, 150 μm thick) and a photocurable sample (RY1101/TMPTA = 7:3, 1 wt% UCPs, 1 wt% BAPO) was subsequently injected into the hole. After 120 s NIR or UV irradiation (Omnicure series1500), the bottom surface of glass sheet was deformed due to the shrinkage stress and then scanned by a 3D line confocal imaging scanner LCI 1220 (LMI TECHNOLOGIES) to characterize the shrinkage stress according to the degree of deformation.

## Tensile testing

Tensile testing was performed with Instron 5967X Universal Testing Systems by using a dumbbell-shaped specimen (length 12 mm, thickness 1 mm, and width 2 mm) at a crosshead speed of 5 mm/min at

25 °C. Three specimens were tested for each sample. Samples for tensile tests were prepared by sandwiching the formulations (RY1101/TMPTA = 7:3, 2 wt% UCPs, 1 wt% BAPO) into a mold made of two glass sheets and a silicone pad, which were then cured under NIR (40 W/cm$^2$ intensity) or UV irradiation (200 mW/cm$^2$ intensity).

## Data availability

All data required to evaluate the conclusions of the dissertation appears in the paper and/or the Supplementary Materials. Other relevant data supporting the findings of this study are available from the corresponding author upon request. Source data are provided with this paper.

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

## Acknowledgements

We thank the financial support by the Science and Technology Support Program of Jiangsu Province (BE2022087, X.S.), the Fundamental Research Funds for the Central Universities and the Postgraduate Research & Practice Innovation Program of Jiangsu Province (KYCX20_1773, P.H.).

## Author contributions

R.L. and X.S. conceived the project; P.H. and R.L. designed the experiments; P.H. conducted all experiments; P.H., H.X. and Y.P. analyzed and interpreted the results; P.H. and R.L. wrote the manuscript, with significant contributions from all authors. All authors approved the final version of the manuscript.

## Competing interests

The authors declare no competing interests.
