## [Peer Review File · Nature Communications]

Upconversion Particle-Assisted NIR Polymerization Enables
Microdomain gradient Photopolymerization at Inter-
particulate Length ScaleReviewers' Comments:

Reviewer #1:

Remarks to the Author:

The article "Insights into Upconversion Particle-Assisted NIR Polymerization: Domain Limited Gradient Photopolymerization at Inter-particulate Length Scale" written by Prof. R. Liu discloses an interesting report about the examination of UCNP cured coatings by radical polymerization and compares this method with traditional UV curing. This team selected several methods to discuss the properties of the cured material and their kinetics in comparison traditional UV curing. The results show a better performance of NIR cured materials which underlines the importance of this curing technique. I am also convinced that research work will give impetus for future research in this field.

Although this manuscript was very carefully written, I still have the following questions/concerns:

1) The discussion about the influence of heat is a bit too weak for me. It was a good idea to increase the heat in the UV experiment. However, this is the macroscopic impression. Authors should discuss more in detail the influence of heat from a microscopic point of view. This contribution should increase significantly higher as 60°C. UCNPs generate about 99% heat. This must be better discussed.

2) The experiment with SiO₂ is not quite clear to me. SiO₂ has another heat capacity which can also affect the overall polymerization process. This experiment is not convincing to me. My proposal is too take it out.

3) Although this group pioneered the field, the article would be benefit by including publications of other groups working in the field. The discussion of results would definitively benefit from this. The reference "Photophysics of Up-Conversion Nanoparticles: Radical Photopolymerization of Multifunctional Methacrylates Comprising Blue- and UV-Sensitive Photoinitiators" ChemPhotoChem 2019, 3, 1119-1126 also disclosed mechanical data of photocured materials, which should be included in the discussion.

Overall, I like the manuscript and recommend publication after fixing of the aforementioned points. It is a very interesting manuscript.

Reviewer #2:

Remarks to the Author:

The present paper of Ren Liu and coworkers describe an interesting approach to reduce shrinkage and shrinkage stress during photopolymerization. The authors use upconverting particles for NIR polymerization. These particles allow a homogenous curing of the material around the particles.

Generally, this is a highly interesting approach and therefore suitable for publication in Nature Communication. The authors use state of the art techniques to characterize the polymerization process very carefully. Also the paper is very well written. I have only two minor comments that have to be considered before this paper is ready for publication.

Line 107: The name of the resin should be mentioned

Figure 7b and corresponding text: Modulus and Tensile toughness should be calculated and shown (at least in the supplementary document)

Reviewer #1

1) The discussion about the influence of heat is a bit too weak for me. It was a good idea to increase the heat in the UV experiment. However, this is the macroscopic impression. Authors should discuss more in detail the influence of heat from a microscopic point of view. This contribution should increase significantly higher as 60°C. UCNPs generate about 99% heat. This must be better discussed.

Absolutely this is a good comment and what we have concerned during UCAP process. The near-infrared optical heating and the exothermic behavior of UCNP will lead to a rapid rise in the temperature during polymerization process. In our previous research, we have investigated the effect of heat on the NIR polymerization process (Laser Induced Thermal Effect on the Polymerization Behavior in Upconversion Particle Assisted Near-Infrared Photopolymerization. *ChemPhysChem*, 2021, 23, 63-68). In the present manuscript, there are two experiments designed for heated UV curing, one is the photorheology test for detecting gel point, and the other is the comparison of shrinkage stress between NIR and UV curing. In the chapter of photorheology test, the reaction temperature for UV curing was determined by monitoring the temperature evolution of monomer under NIR irradiation and got programmed in the rheometer to execute a paralleled UV-curing at high-temperature, we now added it to the Figure S3 (see below). When comparing the gel points between NIR and UV curing systems, the NIR curing condition is 25 W/cm² and 1 wt% UCPs, the equilibrium temperature of monomer under this condition is ~60 °C, so the UV curing system is heated to the same temperature.

In addition, in all curing conditions of the photorheology tests, the maximum temperature could reach 77 °C with NIR intensity of 25 W/cm² and UCNP content of 4wt%, so the gel point test of UV curing at 80 °C is added to Figure S4e (see below), The gel point conversion at this temperature is ~5%, still much lower than NIR curing.

Considering that the equilibrium temperature of the sample at a light intensity of 37 W/cm² (1wt% UCNP) can reach ~80 °C, a shrinkage stress test of the UV-cured sample at 80 °C has been added to the manuscript in Figure 6, see below. the UV-cured material at this temperature showed great shrinkage stress, which is in sharp contrast with the low deformation of NIR curing. All of the above is sufficient to show that the special curing mechanism of UCAP is the main reason for the reduction of shrinkage stress rather than heat.

2)The experiment with SiO₂ is not quite clear to me. SiO₂ has another heat capacity which can also affect the overall polymerization process. This experiment is not convincing to me. My proposal is to take it out.

Thanks for the advice on readability. We took it out from the manuscript, it should be clear now.

3) Although this group pioneered the field, the article would be benefit by including publications of other groups working in the field. The discussion of results would

definitively benefit from this. The reference "Photophysics of Up-Conversion Nanoparticles: Radical Photopolymerization of Multifunctional Methacrylates Comprising Blue- and UV-Sensitive Photoinitiators" ChemPhotoChem 2019, 3, 1119-1126 also disclosed mechanical data of photocured materials, which should be included in the discussion.

The reference you recommended is of great help to our current work. We listed the paper in the Ref. 39 and added relevant discussion in the manuscript.

Reviewer #2:

1)Line 107: The name of the resin should be mentioned

We added the resin name in the mentioned paragraph, this should be clear now.

2)Figure 7b and corresponding text: Modulus and Tensile toughness should be calculated and shown (at least in the supplementary document)

Thanks for reminding us to offer the mechanical properties. We added the Modulus and toughness data in the Table S1 and discussed the data in the manuscript.

Reviewers' Comments:

Reviewer #1:

Remarks to the Author:

My comments were answered accordingly and necessary improvements were made in the revised version.

For my opinion, this manuscript will fit well in this journal.

I recommend publication.

Reviewer #2:

Remarks to the Author:

All the remarks have been addresses well